# Effects of Polymorphism of the Growth Hormone Receptor (GHR) Gene on the Longevity and Milk Performance of White-Backed Cattle

**DOI:** 10.3390/ani15081151

**Published:** 2025-04-17

**Authors:** Karolina Kasprzak-Filipek, Paweł Żółkiewski, Witold Chabuz, Wioletta Sawicka-Zugaj

**Affiliations:** Department of Cattle Breeding and Genetic Resources Conservation, University of Life Sciences in Lublin, Akademicka 13, 20-950 Lublin, Poland; karolina.kasprzak-filipek@up.lublin.pl (K.K.-F.); witold.chabuz@up.lublin.pl (W.C.); wioletta.sawicka@up.lublin.pl (W.S.-Z.)

**Keywords:** growth hormone receptor gene, longevity, herd life, productive life, milking life, white-backed cattle

## Abstract

Longevity in farm animals often refers to productive longevity, determined by productivity and economic outcomes. Functional longevity considers factors beyond productivity, such as health and genetic traits. In dairy cattle, total longevity and functional longevity are important selection parameters. Genetic factors, like the growth hormone receptor (GHR) gene, influence longevity and milk production traits. The Polish White-Backed breed, with unique genetic variants, was studied for GHR gene polymorphism and its association with longevity and milk performance. The study found a strong link between GHR *AluI* locus variation and longevity parameters, suggesting potential applications in breeding for improved longevity and productivity. Further research should include more individuals and breeds to validate these findings.

## 1. Introduction

Longevity in dairy cattle represents one of the most critical yet complex traits influencing the sustainability and profitability of livestock production systems. It is not merely a measure of how long an animal survives, but rather an integrative trait that reflects the interplay between multiple biological, environmental, and management factors. Longevity encompasses both productive life, which measures the duration a cow remains productive within a herd, and functional longevity, which evaluates survival independent of voluntary culling decisions [1]. These distinctions highlight the multifaceted nature of longevity and underscore its importance as a composite trait influenced by production, reproduction, and health attributes. When referring to the lifespan of farm animals, it is most often equated with the period during which the animal is used, while its level of productivity, which translates to economic outcomes, determines whether it will continue to be kept in the herd [2,3]. This is referred to as ‘productive longevity’ [3,4,5,6]. However, it is also worth analyzing what underlies an animal’s potential for long use with an adequate level of productivity. This helps to evaluate what is known as ‘functional longevity’ [2,7]. The literature contains numerous definitions of longevity and diverse methods for estimating it. According to Semenov et al. [8], the productive longevity of cows is one of the most important parameters characterizing the use of the herd. Dairy cattle can be used in this type of analysis owing to the availability of current and archival information on herd management, as well as information on the animals’ origin and health parameters. On the other hand, the life expectancy of individuals is not solely dependent on their health, but on many other factors as well, such as production level and resulting economic considerations [9]. For this reason, separate indicators should be determined in dairy cattle: total longevity, taking into account the actual probability of the animal’s survival, and functional longevity, considering additional factors influencing an animal’s slaughter, irrespective of its productivity and herd management [10]. Functional longevity is thus an important selection parameter in dairy cattle [9]. Numerous studies have attempted to link the length of productive life to other important parameters, such as reproductive parameters [2], udder health [11,12], or the level of inbreeding [2]. The genetic architecture of longevity is inherently intricate, involving numerous biological pathways and processes. Traits related to milk production, reproductive efficiency, and health—such as somatic cell score (a measure of udder health), daughter pregnancy rate (a proxy for fertility), and resistance to metabolic disorders—are genetically correlated with longevity [13]. For instance, cows with superior udder health and fertility are more likely to remain in the herd for extended periods, underscoring the interconnectedness of these traits. However, despite their biological relevance, longevity and its associated traits generally exhibit low heritability, ranging from 0.038 to 0.090 in studies conducted on Holstein populations [14]. This low heritability poses significant challenges for genetic improvement, as it implies that progress through traditional selection methods may be slow and require large datasets for accurate estimation.

According to Szyda et al. [9], however, to learn the determinants of longevity, it is very important to identify the genetic factors that may potentially be associated with it. Candidate genes showing a close relationship with milk production traits may also affect the lifespan of individuals [15,16,17,18]. One such in cattle is the growth hormone receptor (GHR) gene, located on the 20th chromosome [19], and whose 5’UTR region has been studied for its potential influence on milk production indicators [20], carcass traits [21], ligand-binding activity of the liver [22], meat production traits [23], and reproductive performance parameters [24,25].

Given the studies indicating a multifaceted effect of polymorphism at the GHR locus, including its association with lifespan in individuals of different species [26,27,28,29,30], an attempt was made to determine the relationship between polymorphism at the GHR gene locus in cattle and the parameters of longevity. However, the relationship between GHR polymorphisms and longevity remains unclear. Nevertheless, the biological roles of the GHR gene suggest that it may correspond to longevity through effects on health, fertility, and metabolic efficiency. In addition, to extend the analysis, the results obtained for the GHR gene polymorphism were included in relation to milk composition and yield.

The Polish White-Backed (PWB) breed, an indigenous cattle breed covered by a genetic resource conservation programme in Poland, was chosen for the analysis. Our previous research [31] showed that Polish White-Backed cattle, like other indigenous Central European cattle breeds, despite having undergone processes that adversely affected their genetic diversity, remain a reservoir of genetic variants of importance for breeding. A significant factor determining the choice of breed was the fact that among the seven Central European breeds studied, the BB genotype at the GHR gene locus was found only among individuals of the Polish White-Backed breed, which provided the opportunity to determine the relationship between all three genotypes and longevity parameters. In addition, in the light of the latest research [32,33,34,35], as well as current problems with cattle farming, i.e., the significant decrease in the productive life of dairy cows due to early culling, there is a need to identify the genetic factors associated with traits of functionality, health, and longevity in cattle. Therefore, the aim of this study was to investigate the polymorphism at the growth hormone receptor (GHR) gene locus in a population of Polish White-Backed cattle and its association with key parameters of herd and productive life. These parameters included the number of days in milk (milking life), milk yield, and the proportions of individual components in the milk’s dry matter. By identifying genetic factors associated with these traits, this study seeks to provide insights into the genetic determinants of longevity and productivity in cattle, with potential implications for breeding programmes and the long-term sustainability of cattle farming.

## 2. Materials and Methods

### 2.1. Animal Material and Selection Criteria

A total of 100 individuals of the Polish White-Backed (PWB) cattle breed were used in the study. The results of a previous analysis of polymorphism at the GHR gene locus for 50 individuals of the PWB breed were used [31], and taking into account the genetic variation observed in the locus, the research was expanded to include assessment of polymorphism in another 50 individuals.

White-Backed cattle are a restored breed, included in Poland’s genetic resources conservation programme. A herd book for the breed was established in 2003. As of the end of 2023, 1147 cows were registered in the book. During the sampling period for this study (2018–2021), the number of registered cows ranged from 629 to 895. A total of 100 individuals were selected for the study based on the following criteria: documented origin extending at least two generations, dairy utilization, complete life records (including calving dates and drying-off periods, necessary for calculating the analysed indicators), and a minimum of one full lactation. To ensure genetic diversity, only unrelated individuals (verified through breeding documentation and pedigree data of up to two generations) were included. The animals were maintained in a tie-stall system with access to pasture for approximately 8–10 h per day during the spring–autumn seasons. During winter, their diet primarily consisted of homegrown roughage and bruised grain. The farms included in the study typically maintained an average of 10 cows.

### 2.2. Biological Material and Molecular Analysis

The biological material used for the analyses was hair bulbs. Hair was collected from the animals’ necks. DNA was isolated using ready-made commercial kits for the isolation of nucleic acids from biological traces (Sherlock AX A&A Biotechnology, Gdańsk, Poland), according to the procedure recommended by the manufacturer. The analyses aimed at assessing polymorphism in the 5’ flanking region of the growth hormone receptor gene (GHR) were carried out using PCR-RFLP, according to Aggrey et al. [36]. The sequences of the forward and reverse primers were 5′-TGCGTGCACAGCAGCTCAACC-3′ and 5′-AGCAACCCCACTGCTGGGCAT-3′, respectively. The 836 bp PCR product (position −1871 to −1036) was digested with the restriction enzyme *AluI*. The resulting fragments were separated by electrophoresis in a 2% agarose gel. Statistical analysis of the results was carried out using POPGENE 3.2 statistics software.

### 2.3. Longevity and Productivity Data Collection

Polish White-Backed cows from private farms, living in the years 1995–2022, were used in the study (the oldest cow was born in 1995, and the last cow was culled in 2022).

Information on the individual cows was obtained from an ICT system for the registration and evaluation of cows in Poland kept by the Polish Federation of Cattle Breeders and Milk Producers (PFHBiPM, Warsaw, Poland). For analysis of longevity, the following information was obtained from that source: date of birth (DB), date of culling (DCL), date of first calving (DFC), and every subsequent calving (DC), dates of dry periods (DD), and number of lactations (NL). In addition, to evaluate lifetime performance, information on milk yield (MY) was obtained for each complete lactation. For all test-day milking records, data on the content of the main milk ingredients was obtained, i.e., fat (FC), protein (PC), lactose (LC), and dry matter (DMC).

### 2.4. Calculation of Longevity and Productivity Parameters

The data were used to calculate the following parameters using the following formulas:Herd life (days) HL = DCL − DB, where DCL—culling date, DB—date of birthProductive life (days) PL = DCL − DFC, where DCL—culling date, DFC—date of first calvingMilking life (days) ML = PL − ∑DP_1…n_, where PL—productive life, DP—dry period (days), ∑DP_1…n_—sum of dry periods from all lactationsPL to HL ratio (%), where PL—productive life, HL—herd lifeML to PL ratio (%), where ML—milking life, PL—productive lifeLifetime milk yield (kg) LMY = ∑MY_1…n_, where MY—milk yield, ∑MY_1…n_—total milk yield from all lactationsLifetime mean fat content (%) MFC = $\bar{x}$ FC_1…n_, where FC—fat content, $\bar{x}$ FC_1…n_—mean fat content from all lactationsLifetime mean protein content (%) MPC = $\bar{x}$ PC_1…n_, where PC—protein content, $\bar{x}$ PC_1…n_—mean protein content from all lactationsLifetime mean lactose content (%) MLC = $\bar{x}$ LC_1…n_, where LC—lactose content, $\bar{x}$ LC_1…n_—mean lactose content from all lactationsLifetime mean dry matter content (%) MDMC = $\bar{x}$ DMC_1…n_, where DMC—dry matter content, $\bar{x}$ DMC_1…n_—mean dry matter content from all lactations.

The nomenclature of the main parameters was taken from Schuster et al. [37] and Hu et al. [3].

### 2.5. Statistical Analysis

The significance of differences for means was tested using one-way analysis of variance (ANOVA) and Tukey’s multiple range. The computations were carried out using STATISTICA 13.0 software. Significance was determined for highly significant differences at *p* ≤ 0.01 (A and B) and for significant differences at *p* ≤ 0.05 (a and b).

## 3. Results

The analysed GHR *AluI* polymorphism is the result of a mutation of the transversion type (A to T) at position −1182. Restriction digestion allowed us to obtain 3 genotypes: AA (with restriction fragment lengths: 747 bp and 75 bp), AB (747 bp, 602 bp, 145 bp, 75 bp), and BB (602 bp, 145 bp, and 75 bp, respectively). The frequencies were 0.60, 0.32, and 0.08, respectively, with allele frequencies of 0.76 for GHRA and 0.24 for GHRB (Table 1).

Taking into account the distribution of frequencies in the individual farms (Table 2), it should be noted that the AA genotype dominated in most of them. It was the only one found in the case of two farms (7 and 8). Notably, farm 10 was dominated by the AB genotype. Homozygotes BB were found in 50% of all the farms analysed.

Data collected in the ICT system kept by PFHBiPM, where all information on the use and breeding of dairy cattle in Poland is stored, were used to estimate herd, productive, and milking life (Table 3). These parameters were calculated for each genotype and expressed in days. For all three parameters, statistically highly significant (*p* ≤ 0.01) differences between the AA and AB genotypes and the BB genotype were observed. Homozygous individuals for allele A at the GHR locus had the highest values for the longevity parameters analysed, i.e., the most total days from birth to culling (herd life), the longest productive life, estimated as the number of days from the first calving to culling, and the longest milking life (highest total number of days in milk), calculated as production periods minus dry periods.

The mean HL of the Polish White-Backed cows was 10.5 years (3849.3 days). The highest values for this parameter were noted in the group of individuals with the AA genotype (on average 4091.5 days, i.e. 11.2 years) (Table 3), and the cow with the highest total lifespan among all those analysed lived 6291 days, i.e., 17.2 years.

The results show that the average PL of the cows was 2988.9 days, i.e., 8.2 years. Individuals with the AA genotype were used longest (on average 3258.4 days, i.e., 8.9 years), while individuals with the BB genotype had the shortest productive life (on average 1079.0 days, i.e., 3 years). These differences were statistically highly significant (*p* ≤ 0.01) between the AA and AB genotypes and the BB genotype. The number of lactations was also highly varied between the genotype groups. The highest mean number of lactations was recorded for individuals with the AA genotype and the lowest for the BB genotype, which is confirmed by the statistically highly significant differences between these groups. The cow with the most years of use (15 years) and the most lactations (14) had the AA GHR *AluI* genotype.

The ratios of productive life to herd life (PL/HL) were lowest in individuals with the BB genotype, which had the shortest total herd life. No statistically significant differences were noted in the last of the parameters included in Table 3, i.e., the ratio of milking life to productive life (ML/PL); this parameter was highest for individuals with the BB genotype, but the greatest differences between individuals were noted in this group, as indicated by the standard deviation (SD).

The data presented in Table 4 show that the parameter of lifetime milk yield among BB individuals varied highly significantly (*p* ≤ 0.01) compared to AB and AA individuals.

The average value for the yield in relation to the length of herd life parameter was lowest in the group of homozygous individuals for the B allele and differed highly significantly (*p* ≤ 0.01) from the values estimated for individuals with the AB and AA genotypes. For the last of the parameters in Table 4, i.e., yield per day of milking life, individuals with AA genotypes had the lowest average values, which differed significantly statistically (*p* ≤ 0.05) from those obtained for individuals with the AB and BB genotypes.

The analyses carried out to evaluate the relationship between the genotype at the GHR *AluI* gene locus and the parameters presented in Table 5 revealed that only lactose content was associated with this polymorphism.

Regarding the percentage share (%) of parameters in Table 5, despite the statistically non-significant differences, it is worth noting that the highest values were obtained among individuals with the BB genotype, which indicated more favourable milk quality parameters and better suitability for processing due to the higher percentages of dry matter ingredients.

## 4. Discussion

The frequency of the A allele obtained in this study was higher than that estimated by Aggrey et al. [36] in a study conducted in two groups of Holstein bulls born in 1950–1970 and in the 1980s. The frequencies of the A (*AluI* (−)) allele were 0.63 and 0.42, respectively. Lower frequencies of the A allele than those obtained among Polish White-Backed cattle were also reported by Skinkyte et al. [38] for individuals of the Lithuanian Red and Lithuanian Black-and-White breeds: 0.36 and 0.55, respectively. Similarly, in a study by Rahbar et al. [39], the frequency of the A (*AluI* (−)) allele was lower than in the present study, amounting to 0.56. In contrast, Deepika and Salar [40], in a study of individuals of indigenous Indian cattle breeds, reported a higher frequency of the A allele. They found that the frequency of the A allele for 10 breeds ranged from 0.833 to 0.968. Also, Olenski et al. [41] estimated a higher frequency of the A allele for Polish Holstein Friesian (PHF) cattle (GHRA 0.832).

The Polish White-Backed breed was chosen for further analysis because three genotypes occurred in the GHR *AluI* locus in these individuals. Previous analyses have shown that the BB genotype was not present in Central European cattle [31]. In this study, when the research material was expanded to include more individuals of the Polish White-Backed breed, the frequency of the BB genotype was estimated at 0.08. Contrasting results were obtained by Skinkyte et al. [38] in populations of the Lithuanian Black-and-White and Lithuanian Red breeds; the frequency of the BB genotype was higher, at 0.15 and 0.43, respectively.

The results indicate that the average length of productive life of the cows included in the study was 8.2 years (i.e., 2988.9 days). Available results of studies by other authors indicate that this parameter had high values in Polish White-Backed cows. In a study by Chirinos et al. [42] conducted in a Spanish population of HF cattle, the length of productive life (PL) ranged from 828 to 1059 days. Similarly, lower PL values for individuals of the Slovenian Brown cattle breed were reported by Jenko et al. [43], with a median value of 1192 days. Pachova et al. [44] also reported a low value for this parameter, on average 923 days. Similarly, Strapakova et al. [7] obtained PL values of 898 days for HF cattle and 985 days for Slovak Simmental cattle. Studies by Morek-Kopeć and Zarnecki [45,46] showed that in Polish populations of Simmental and Holstein–Friesian cattle, the average length of productive life was 1198 days and 936.6 days, respectively. A study by Gnyp [47] on a population of Polish Black-and-White cattle showed that the productive life of these cows ranged from 1275 to 1251 days.

According to Rostellato et al. [34] and Hu et al. [3], the biological lifespan of cows is about 20 years. In practice, the length of a cow’s productive life is much shorter than that determined by biological factors [48]. The high herd values obtained in Polish White-Backed individuals show that cows of this breed have high potential for long use. Due to breed-specific management strategies—such as a feeding system that promotes healthier animals (primarily based on pasture or homegrown fodder) and a dual-purpose maintenance system that does not prioritize maximum milk yield—Polish White-Backed cattle can be maintained for a significantly longer period. The milk produced by this breed is characterized by a high protein and fat content, further supporting its value in sustainable farming. An important factor contributing to the extended longevity of White-Backed cattle is the availability of subsidies for breeders, which help compensate for lower milk yields while supporting breed conservation efforts. Additionally, the lack of selection for high milk production reduces the incidence of diseases commonly associated with high-yielding cows, further enhancing the breed’s longevity. Compared with available data, the values obtained for White-Backed cattle lifespan are high. Analyses conducted by Joshi et al. [33] among individuals of the Indian breed Kankrej showed that the average lifespan (expressed as the number of days from the animal’s birth to its sale or death) was 2568.72 days (about 7 years). Similarly, studies by other authors have obtained values lower than those recorded in the present study. Gnyp [47] reported a herd life of 2125–2099 days (about 5.8 years) in Polish Black-and-White cows, while Hu et al. [3] estimated that the average productive life of cows in a Chinese HF population was 850.99 days (about 2.3 years).

The parameter of average productive life is linked to the number of lactations. It is the number of lactations that determines the individual’s total lifetime milk yield (LMY), which translates into economic outcomes of the herd [47]. For this reason, according to Miciński [49], Sawa [50], and Sobek et al. [51], taking into account the profitability of production, dairy cows should be used for at least six or seven lactations. If they are culled too early, e.g., due to low yield, they do not achieve peak production, which usually occurs in the third or fourth lactation [52]. According to Imbayarwo-Chikosi et al. [53], cows with an average milk yield have the longest productive lives. The findings reported by Chirinos et al. [42] also indicate a very significant relationship between milk yield and length of productive life. The results of the conducted study confirm that individuals with the AB genotype, which had the highest lifetime milk yield (41,202.8 kg), were used for a shorter time (8.1 years; 7.2 lactations) than individuals with the AA genotype, which had a somewhat lower lifetime milk yield (39,430.5 kg) and were used longer (8.9 years; 7.3 lactations). Lifetime milk yield was at the same time more than 10,000 kg higher than that noted in other indigenous European breeds [54].

Another parameter indicating performance efficiency is milk yield in relation to the length of the cow’s herd and productive life. For the first of these, White-Backed cows with the AA and AB genotypes had values similar to those reported for PHF [55]. The values for yield in relation to productive life were lower due to the much higher number of lactations, and thus more dry periods of greater length over the course of a lifetime. Heterozygous individuals also had a higher yield per day of milking life, i.e., 17.0 kg. This value is lower than in PHF, reported as 19.5–21.9 kg [56], but their milking life was nearly twice as long.

With regard to the individual milk ingredients (Table 5), these parameters can be seen to be closely linked to both external factors, such as the time of year, humidity, temperature, and resulting heat stress [57,58], and internal factors, such as the stage of lactation and the animal’s health or body condition [59,60]. According to Rahbar et al. [39], a very important factor associated with the development of the mammary gland and milk traits is variation in the growth hormone receptor (GHR) gene. The authors reported that milk from GHR *AluI* (+) cows contained more protein and fat in the first lactation than from *AluI* (−). Aggrey et al. [36], who analysed the relationship between polymorphism in the 5’ flanking region of the growth hormone receptor gene in HF bulls and the milk traits of their daughters, also found a very strong link between this polymorphism and the protein content of the milk. Cobanoglu et al. [61], in a study conducted in a Turkish dairy cattle population, also emphasized that the GHR gene shows a relationship with milk characteristics and should be considered as a candidate gene in selection programmes aimed at improving milk yield and milk quality parameters.

## 5. Conclusions

This study reported the association between the GHR *AluI* polymorphism and some longevity parameters, including herd life, productive life, and milking life in the Polish White-Backed cattle population. Additionally, the study found that this polymorphism was linked to lifetime milk yield. These findings suggest that genetic markers, such as the GHR *AluI* polymorphism, could be valuable tools in breeding programmes aimed at enhancing cattle productivity and longevity. The study also emphasizes the importance of conserving and breeding native cattle breeds to maintain their unique genetic traits, ensuring the sustainability of cattle farming in the future.

## Figures and Tables

**Table 1 animals-15-01151-t001:** Numbers and frequencies of genotypes and alleles at the GHR *AluI* locus.

GHR *AluI*	Genotype			Alleles	
	AA	AB	BB	A	B
Frequency	0.6	0.32	0.08	0.76	0.24
Number of animals	60	32	8		

**Table 2 animals-15-01151-t002:** Allele frequencies and genotype numbers in the farms studied.

Farm	Number of Animals in Each Genotype			Allele Frequency	
	AA	AB	BB	A	B
1	6	2	-	0.87	0.13
2	6	3	2	0.68	0.32
3	9	2	2	0.77	0.23
4	5	4	1	0.70	0.30
5	7	-	-	1.00	-
6	8	-	-	1.00	-
7	6	2	-	0.87	0.13
8	6	5	2	0.65	0.35
9	5	3	-	0.81	0.19
10	2	11	1	0.54	0.46

**Table 3 animals-15-01151-t003:** Longevity parameters in relation to the genotype at the GHR *AluI* locus.

GHR *AluI* Genotype		AA	AB	BB	$\bar{x}$
Herd Life—HL (days)	$\bar{x}$	4091.5 A	3836.4 A	2084.0 B	3849.3
	SD	1089.1	926.2	368.5	1128.1
Productive life—PL (days)	$\bar{x}$	3258.4 A	2961.0 A	1079.0 B	2988.9
	SD	1090.0	931.8	523.8	1156.9
Milking life—ML (days)	$\bar{x}$	2614.7 A	2395.7 A	912.0 B	2408.4
	SD	803.4	650.3	490.9	860.8
Number of lactations—NL	$\bar{x}$	7.3 A	7.2 A	2.3 B	7.0
	SD	2.6	2.3	0.6	2.7
PL/HL	$\bar{x}$	78.03 A	75.70 A	49.47 B	75.00
	SD	7.75	7.67	18.79	11.73
ML/PL	$\bar{x}$	81.49	82.30	82.38	81.82
	SD	7.26	6.43	10.41	7.22

A,B—means in rows with different letters are significantly different (*p* ≤ 0.01).

**Table 4 animals-15-01151-t004:** Lifetime milk yield in relation to the longevity of cows depending on the genotype at the GHR *AluI* locus.

GHR *AluI* Genotype		AA	AB	BB	$\bar{x}$
Lifetime milk yield (kg)	$\bar{x}$	39,430.5 A	41,202.8 A	12,072.0 B	37,809.0
	SD	13,540.8	15,562.3	3680.2	15,649.5
Yield/day of herd life	$\bar{x}$	9.7 A	10.6 A	5.7 B	9.7
	SD	2.3	2.5	1.2	2.6
Yield/day of productive life	$\bar{x}$	12.6	13.9	13.2	13.1
	SD	3.3	3.0	5.5	3.4
Yield/day of milking life	$\bar{x}$	15.3 a	17.0 b	16.8 b	16.0
	SD	3.4	3.7	8.9	4.2

a,b—means in rows with different letters are significantly different (*p* ≤ 0.05). A,B—means in rows with different letters are significantly different (*p* ≤ 0.01).

**Table 5 animals-15-01151-t005:** Percentages of major milk ingredients in relation to the genotype at the GHR *AluI* locus.

GHR *AluI* Genotype		AA	AB	BB	$\bar{x}$
Average lifetime fat content (%)	$\bar{x}$	3.84	4.00	4.05	3.91
	SD	0.42	0.47	0.17	0.43
Average lifetime protein content (%)	$\bar{x}$	3.26	3.21	3.31	3.25
	SD	0.28	0.20	0.11	0.25
Average lifetime lactose content (%)	$\bar{x}$	4.63 B	4.68	4.77 A	4.66
	SD	0.16	0.12	0.06	0.15
Average lifetime dry matter content (%)	$\bar{x}$	12.42	12.60	12.80	12.51
	SD	0.56	0.65	0.19	0.58

A,B—means in rows with different letters are significantly different (*p* ≤ 0.01).

## Data Availability

The datasets used and/or analysed during the current study are available from the corresponding author upon reasonable request.

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
