# Peer review of "Effects of Polymorphism of the Growth Hormone Receptor (GHR) Gene on the Longevity and Milk Performance of White-Backed Cattle"

_animals, 2025, doi:10.3390/ani15081151_

Round 1

Reviewer 1 Report

Comments and Suggestions for Authors

The manuscript examines the association between polymorphisms in the growth hormone receptor (GHR) gene and key parameters related to longevity and milk production in Polish White-Backed cattle. While the manuscript is generally well structured and clearly written, some stylistic and grammatical issues are present.

Certain sections contain overly complex sentence structures, making them difficult to follow.

Lines 52–73: contain unnecessarily detailed information on the GH, INS, IGF pathway, which could be condensed for clarity without losing essential information.

A key concern is the lack of clarity regarding the origin of the animals used for genotyping. It is unclear whether the animals were from the same or different herds, which could introduce environmental bias. The authors should provide a detailed description of the sampled animals, including whether the samples were collected in the same year or over multiple years. Additionally, it is important to specify how environmental variation was accounted for in the analysis.

The Discussion section is generally well organized, but some areas may need improvement.

Lines 305–307: The statement "cows of this breed have high potential for long use" is too general. Therefore, it would be beneficial to specify breed-specific management strategies that contribute to longevity.

Address the limitations of the study, particularly the sample size (100 individuals), which limits the strength of population-level conclusions. Additionally, the low frequency of some genotypes (due to the small population size) further constrains the ability to draw general conclusions.

There is no explicit mention of how environmental factors influencing longevity and milk production were controlled. The authors should discuss how potential confounding variables—such as nutrition, farm management, and disease might have affected the longevity. Providing a more detailed discussion of the potential confounding factors would significantly strengthen the conclusions of this study.

Comments on the Quality of English Language

The manuscript is generally well written, but some stylistic and grammatical issues are present. 

Author Response

At the outset, thank you for the review. The valuable comments and suggestions included in it will assist us in improving our article.

Comments 1: The manuscript examines the association between polymorphisms in the growth hormone receptor (GHR) gene and key parameters related to longevity and milk production in Polish White-Backed cattle. While the manuscript is generally well structured and clearly written, some stylistic and grammatical issues are present.

Response 1: The manuscript has been re-checked by a native speaker, and the changes made have been marked in the manuscript. However, since no specific line references were provided, we cannot be certain that all of the reviewer's suggested corrections have been included.

Comments 2: Certain sections contain overly complex sentence structures, making them difficult to follow.

Response 2: Thank you for pointing out the potential difficulty in following certain sections of the manuscript. We have revised the text by shortening overly complex sentences to improve clarity (Lines 53-93).

Comments 3: Lines 52–73: contain unnecessarily detailed information on the GH, INS, IGF pathway, which could be condensed for clarity without losing essential information.

Response 3: According to the authors, the information on the GH, INS, and IGF pathway is closely related to the growth hormone receptor and their description is intended to explain the relationship of the above-mentioned hormones to the lifespan of both low-organized organisms and mammals. This underscores the complexity of factors influencing lifespan, as well as explains the conserved nature of this mechanism. Therefore, simplifying this section would deprive the manuscript of a valuable background. However, in response to the reviewer’s suggestion, the authors have made minor revisions to simplify overly complex phrases.

Comments 4: A key concern is the lack of clarity regarding the origin of the animals used for genotyping. It is unclear whether the animals were from the same or different herds, which could introduce environmental bias. The authors should provide a detailed description of the sampled animals, including whether the samples were collected in the same year or over multiple years. Additionally, it is important to specify how environmental variation was accounted for in the analysis.

Response 4: In response to the reviewer’s suggestion, changes have been made to the manuscript (Lines 148-162). As explained, the animals came from different farms but were maintained under the same animal housing system: tie-stall barns with grazing during the spring-autumn season and feeding based on their own fodder during the winter. Given the similar maintenance and feeding conditions, environmental factors were not considered as differentiating variables in the study.

Comments 5: The Discussion section is generally well organized, but some areas may need improvement.

Response 5: The suggested content from the reviewer has been added into the Discussion section, specifically regarding the factors influencing the longevity of Polish White-Backed cattle.

Comments 6: Lines 305–307: The statement "cows of this breed have high potential for long use" is too general. Therefore, it would be beneficial to specify breed-specific management strategies that contribute to longevity.

Response 6: The authors agree with the reviewer's suggestion regarding the generality of the statement. In the revised manuscript (Lines 331-342​), this statement has been revised, and breed-specific management strategies contributing to longevity are now described.

Comments 7: Address the limitations of the study, particularly the sample size (100 individuals), which limits the strength of population-level conclusions. Additionally, the low frequency of some genotypes (due to the small population size) further constrains the ability to draw general conclusions.

Response 7: In the Materials and Methods section (Lines 148-158​), we have addressed the criteria used for selecting animals for the study. As noted in the manuscript, the Polish White-Backed breed is a restored breed and is part of a genetic resource conservation program. In the last year of sample collection (2021), the total number of cows in the herd book was 895 (currently 1,147). Initially, the entire existing population was considered for the study. However, after applying the selection criteria (as described in the manuscript), individuals who did not meet the criteria were excluded, resulting in a final sample of about 100 individuals. This represented approximately 11% of the entire population and nearly 100% of those meeting the established criteria. Therefore, despite the low frequency of some genotypes, the group of 100 individuals represents the largest possible sample based on the selection criteria, and the conclusions drawn are intended to apply to the entire population.

Comments 8: There is no explicit mention of how environmental factors influencing longevity and milk production were controlled. The authors should discuss how potential confounding variables—such as nutrition, farm management, and disease might have affected the longevity. Providing a more detailed discussion of the potential confounding factors would significantly strengthen the conclusions of this study.

Response 8: As mentioned in the manuscript (Lines 158-162), environmental factors across all farms were very similar, and therefore, they were not considered as a differentiating factor in the study. In Lines 331-342, we further discuss the breed management practices that explain how factors such as nutrition, maintenance system, and disease affect longevity. Specifically, we address the lack of a focus on high milk yield, which leads to a lower frequency of diseases typically found in high-yielding animals; the subsidies provided to breeders that encourage long-term use of animals; and the management system that involves pasture feeding, which promotes better animal health and, consequently, longer productive lifespans.

Comments 9: The manuscript is generally well written, but some stylistic and grammatical issues are present.

Response 9: The manuscript has been re-checked by a native speaker, and stylistic and grammatical revisions have been made accordingly. All changes have been marked in the manuscript.

Reviewer 2 Report

Comments and Suggestions for Authors

Material and Methods: Given the relationship between the variables and their association with the genotypes, Did the authors consider applying a multivariate linear regression model to the analysis?

Lines 253 – 258: If the analysis is based on proportions, no statistical difference would be expected. Have the authors considered analyzing these parameters as total production during productive life, i.e., the total kg of fat, protein, etc., throughout productive life, and their relationship with genotypes?

Lines 204, 206, 207 …: On lines 195 - 196 it is clarified that the values ​​are expressed in days, it is not necessary to specify in each parameter whether it is in days or years. Homogenize the units, please.

The authors conclude that there is a strong association between variation at the GHR AluI locus with the parameters evaluated (lines 362 and 363) so they should rethink the objective in the introduction section to exclude the verb “attempt”

Author Response

At the outset, thank you for the review. The valuable comments and suggestions included in it will assist us in improving our article.

Comments 1. Material and Methods: Given the relationship between the variables and their association with the genotypes, did the authors consider applying a multivariate linear regression model to the analysis?

Response 1. We performed a multivariate linear regression analysis, considering the GHR genotype, year of birth, age at first calving, herd effect, and culling reason as independent variables. The results showed a moderate strength in the linear regression model's ability to explain the variability in the dependent variables. The R-squared and Adjusted R-squared values indicated that the model accounted for between 4% and 36% of the variability, depending on the specific dependent variable. In most cases, the GHR genotype had a significant impact on the dependent variables, similar to the findings from the ANOVA analysis. Since the animals were maintained on farms with similar management and feeding systems (as detailed in the response to Reviewer 1, Lines: 158-162), it was challenging to include additional environmental factors into the model.

Comments 2. Lines 253 – 258: If the analysis is based on proportions, no statistical difference would be expected. Have the authors considered analyzing these parameters as total production during productive life, i.e., the total kg of fat, protein, etc., throughout productive life, and their relationship with genotypes?

Response 2. In the initial draft of the manuscript, we analyzed the lifetime yield of individual milk components. However, we decided not to present these results, as they depend on both total lifetime milk yield and component content.

Comments 3. Lines 204, 206, 207 …: On lines 195 - 196 it is clarified that the values are expressed in days, it is not necessary to specify in each parameter whether it is in days or years. Homogenize the units, please.

Response 3. The use of both days and years follows the conventions used in various publications and allows for a more precise presentation of results. We believe that using dual units enhances the visualization of the findings and facilitates comparisons with other studies.

Comments 4. The authors conclude that there is a strong association between variation at the GHR AluI locus with the parameters evaluated (lines 362 and 363) so they should rethink the objective in the introduction section to exclude the verb “attempt.”

Response 4. In accordance with the reviewer's suggestion, this part of the manuscript has been revised (Lines 129-137).

Reviewer 3 Report

Comments and Suggestions for Authors

The manuscript entitled, “Effects of polymorphism of the growth hormone receptor (GHR) gene on the longevity and milk performance of white-backed cattle” clearly demonstrates the inter-relationship between polymorphism in GHR gene locus in Polish White-backed cattle population and significant longevity parameters. Furthermore, the results proved that the polymorphism was linked to the level of milk yield and content of individual dry matter constituents of milk. The proposed study results can aid in strategizing future breeding programs ensuring economic security of cattle producers. In my opinion, the manuscript has high novelty and is authored very well. But before the paper is considered for publication the authors must address the following points:

 Introduction

  • Consider revising the font in line 60, “Saccharomyces cerevisiae”
  • Forty four references in this section is overloading, consider simmering the contents from line 74-111. This section as a whole can be summarised and re-organized
  • Line 126 to 130, the research objective and hypothesis can be refined

Materials and method

  • In this section, the method of collection of the biological samples from study animals can be included
  • Information about the primer source/the designing software can be mentioned

Results

  • Line 193, “ITC system”, check the spelling errors

Discussion

  • In general, the discussion can be improved as it includes only study comparisons. Many results (line 262-299) are presented as such without any scientific explanation, making it unconvincing. The possible reasons can be summarised for better comprehension

Conclusion

  • The limitations of the study can be presented

Author Response

At the outset, thank you for the review. The valuable comments and suggestions included in it will assist us in improving our article.

Comments 1: Consider revising the font in line 60, “Saccharomyces cerevisiae.”

Response 1: The font for Saccharomyces cerevisiae has been corrected as per the reviewer's suggestion

Comments 2: Forty-four references in this section are overloading. Consider summarizing the contents from lines 74-111. This section as a whole can be summarized and reorganized.

Response 2:  Thank you for your suggestion. We have revised the manuscript to simplify the text in this section. However, due to the critical role of this section in standardizing the terms used in the study, we believe that retaining the citations is necessary and have therefore not made further modifications to the references.

Comments 3: Lines 126 to 130, the research objective and hypothesis can be refined.

Response 3: This section of the manuscript has been revised to better clarify both the scientific and practical objectives of the study (Lines 129-137).

Comments 4: In this section, the method of collection of the biological samples from study animals can be included.

Response 4: The "Material and Methods" section (Line 163) there is the information on the biological sample collection method. However, in response to the reviewer's suggestion, we have provided additional details to further clarify the procedure.

Comments 5: Information about the primer source/the designing software can be mentioned.

Response 5: Information about the primer source is provided in Lines 168-170 of the manuscript.

Comments 6: Line 193, “ITC system,” check the spelling errors.

Response 6: The spelling error has been corrected.

Comments 7: In general, the discussion can be improved as it includes only study comparisons. Many results (lines 262-299) are presented as such without any scientific explanation, making it unconvincing. The possible reasons can be summarized for better comprehension.

Response 7: Revisions have been made to this section of the manuscript to include explanations of possible connections between various factors and the longevity of the White-Backed breed (Lines 331-342). These changes aim to provide a more comprehensive and scientifically grounded discussion.

Comments 8: The limitations of the study can be presented.

Response 8: Information regarding the study's limitations, particularly in the context of future research in this field, has been included in the revised summary section (Lines 398-416).

Reviewer 4 Report

Comments and Suggestions for Authors

Independent review animals-3513149

Article type: research article.

Title: Effects of polymorphism of the growth hormone receptor (GHR) gene on the longevity and milk performance of white-backed cattle

The research article evaluates polymorphism in the GHR gene locus in relation to longevity, productivity parameters, and the content of individual components of milk dry matter of white-backed cattle. The study identified three genotypes: AA, AB, and BB, each with different frequencies. Additionally, the study found that polymorphism in the GHR gene locus is strongly associated with cows' longevity and days in milk. The number of animals studied is sufficient (n=100); the study is well designed, the statistics are appropriate, the results are clearly presented, and the discussion aligns with the results. I can conclude from this study the importance of raising indigenous cattle herds with pure breeding because they are a valuable genetic resource.  However, I have some concerns: Line 6 - 14: I noticed that all authors of the paper share the same affiliation. Therefore, please use only one affiliation for all authors.

Line 84-86: Please rephrase that sentence for more clarity.

Suggestion: Dairy cattle can be used in this type of analysis due to the availability of current and archival information on herd management, as well as information on the animals' origin and health parameters.

Line 126-130: Please revise the following sentence to clearly highlight the impact of the study's objectives.

Line 295: Please delete the supplementary space.

Line 358-367: Please ensure that the conclusions section includes a sentence highlighting the importance of raising these indigenous cows in pure breed in the future.

Comments on the Quality of English Language

Some sentences could be revised for clarity to better present the objectives of the study.

Author Response

At the outset, thank you for the review. The valuable comments and suggestions included in it will assist us in improving our article.

Comments 1: Line 6 - 14: I noticed that all authors of the paper share the same affiliation. Therefore, please use only one affiliation for all authors.

Response 1: The use of separate affiliation links is due to the different email addresses of the authors. In the view of the authors, the template of the paper suggests this way of affiliation format.

Comments 2: Line 84-86: Please rephrase that sentence for more clarity.

Response 2: Thank you for drawing attention to this sentence. It has been corrected in the manuscript.

Comments 3: Line 126-130: Please revise the following sentence to clearly highlight the impact of the study's objectives.

Response 3: As the reviewer suggested, the purpose of the paper was modified (Lines 129-137).

Comments 4: Line 295: Please delete the supplementary space.

Response 4: Thank you for pointing out this obvious mistake. It has been included in the revised version of the manuscript. 

Comments 5: Line 358-367: Please ensure that the conclusions section includes a sentence highlighting the importance of raising these indigenous cows in pure breed in the future.

Response 5: The conclusions section has been revised and the modified version emphasizes the importance of raising indigenous cattle in pure breed in the future, as a valuable reservoir of genes responsible, for example, for functional traits such as longevity (Lines 398-416).

Round 2

Reviewer 3 Report

Comments and Suggestions for Authors

The authors have addressed all my queries satisfactorily 

Author Response

Thank you for your review.